# Faster Linear Algebra for Distance Matrices

**Piotr Indyk**
MIT
indyk@mit.edu

**Sandeep Silwal**
MIT
silwal@mit.edu

## Abstract

The distance matrix of a dataset $X$ of $n$ points with respect to a distance function $f$ represents all pairwise distances between points in $X$ induced by $f$. Due to their wide applicability, distance matrices and related families of matrices have been the focus of many recent algorithmic works. We continue this line of research and take a broad view of algorithm design for distance matrices with the goal of designing fast algorithms, which are specifically tailored for distance matrices, for fundamental linear algebraic primitives. Our results include efficient algorithms for computing matrix-vector products for a wide class of distance matrices, such as the $\ell_1$ metric for which we get a linear runtime, as well as an $\Omega(n^2)$ lower bound for any algorithm which computes a matrix-vector product for the $\ell_\infty$ case, showing a separation between the $\ell_1$ and the $\ell_\infty$ metrics. Our upper bound results, in conjunction with recent works on the matrix-vector query model, have many further downstream applications, including the fastest algorithm for computing a relative error low-rank approximation for the distance matrix induced by $\ell_1$ and $\ell_2^2$ functions and the fastest algorithm for computing an additive error low-rank approximation for the $\ell_2$ metric, in addition to applications for fast matrix multiplication among others. We also give algorithms for constructing distance matrices and show that one can construct an approximate $\ell_2$ distance matrix in time faster than the bound implied by the Johnson-Lindenstrauss lemma.

## 1 Introduction

Given a set of $n$ points $X = \{x_1, \ldots, x_n\}$, the distance matrix of $X$ with respect to a distance function $f$ is defined as the $n \times n$ matrix $A$ satisfying $A_{i,j} = f(x_i, x_j)$. Distances matrices are ubiquitous objects arising in various applications ranging from learning image manifolds [TSL00, WS06], signal processing [SY07], biological analysis [HS93], and non-linear dimensionality reduction [Kru64, Kru78, TSL00, CC08], to name a few[1]. Unfortunately, explicitly computing and storing $A$ requires at least $\Omega(n^2)$ time and space. Such complexities are prohibitive for scaling to large datasets.

A silver lining is that in many settings, the matrix $A$ is not explicitly required. Indeed in many applications, it suffices to compute some underlying function or property of $A$, such as the eigenvalues and eigenvectors of $A$ or a low-rank approximation of $A$. Thus an algorithm designer can hope to use the special geometric structure encoded by $A$ to design faster algorithms tailored for such tasks.

Therefore, it is not surprising that many recent works explicitly take advantage of the underlying geometric structure of distance matrices, and other related families of matrices, to design fast algorithms (see Section 1.2 for a thorough discussion of prior works). In this work, we continue this line of research and take a broad view of algorithm design for distance matrices. Our main motivating question is the following:

---

[1]We refer the reader to the survey [DPRV15] for a more thorough discussion of applications of distance matrices.

36th Conference on Neural Information Processing Systems (NeurIPS 2022).

*Can we design algorithms for fundamental linear algebraic primitives which are
specifically tailored for distance matrices and related families of matrices?*

We make progress towards the motivating question by studying three of the most fundamental
primitives in algorithmic linear algebra. Specifically:

1. We study upper and lower bounds for computing matrix-vector products for a wide array of
   distance matrices,

2. We give algorithms for multiplying distance matrices faster than general matrices, and,

3. We give fast algorithms for constructing distance matrices.

## 1.1 Our Results

We now describe our contributions in more detail.

*1. We study upper and lower bounds for constructing matrix-vector queries for a
wide array of distance matrices.*

A matrix-vector query algorithm accepts a vector $z$ as input and outputs the vector $Az$. There is
substantial motivation for studying such queries. Indeed, there is now a rich literature for fundamental
linear algebra algorithms which are in the "matrix free" or "implicit" model. These algorithms only
assume access to the underlying matrix via matrix-vector queries. Some well known algorithms in
this model include the power method for computing eigenvalues and the conjugate gradient descent
method for solving a system of linear equations. For many fundamental functions of $A$, nearly
optimal bounds in terms of the number of queries have been achieved [MM15, BHSW20, BCW22].
Furthermore, having access to matrix-vector queries also allows the simulation of any randomized
sketching algorithm, a well studied algorithmic paradigm in its own right [Woo14]. This is because
randomized sketching algorithms operate on the matrix $\Pi A$ or $A\Pi$ where $\Pi$ is a suitably chosen
random matrix, such as a Gaussian matrix. Typically, $\Pi$ is chosen so that the sketches $\Pi A$ or $A\Pi$
have significantly smaller row or column dimension compared to $A$. If $A$ is symmetric, we can easily
acquire both types of matrix sketches via a small number of matrix-vector queries.

Therefore, creating efficient versions of matrix-vector queries for distance matrices automatically
lends itself to many further downstream applications. We remark that our algorithms can access to
the set of input points but *do not* explicitly create the distance matrix. A canonical example of our
upper bound results is the construction of matrix-vector queries for the function $f(x, y) = \|x - y\|_p^p$.

**Theorem 1.1.** *Let $p \geq 1$ be an integer. Suppose we are given a dataset of $n$ points $X = \{x_1, \ldots, x_n\} \subset \mathbb{R}^d$. $X$ implicitly defines the matrix $A_{i,j} = \|x_i - x_j\|_p^p$. Given a query $z \in \mathbb{R}^n$, we
can compute $Az$ exactly in time $O(ndp)$. If $p$ is odd, we also require $O(nd \log n)$ preprocessing time.*

We give similar guarantees for a wide array of functions $f$ and we refer the reader to Table 1 which
summarizes our matrix-vector query upper bound results. Note that some of the functions $f$ we
study in Table 1 do not necessarily induce a metric in the strict mathematical sense (for example the
function $f(x, y) = \|x - y\|_2^2$ does not satisfy the triangle inequality). Nevertheless, we still refer to
such functions under the broad umbrella term of "distance functions" for ease of notation. We always
explicitly state the function $f$ we are referring to.

Crucially, most of our bounds have a linear dependency on $n$ which allows for scalable computation
as the size of the dataset $X$ grows. Our upper bounds are optimal in many cases, see Theorem A.13.

Combining our upper bound results with optimized matrix-free methods, immediate corollaries of
our results include faster algorithms for eigenvalue and singular value computations and low-rank
approximations. Low-rank approximation is of special interest as it has been widely studied for
distance matrices; for low-rank approximation, our bounds outperform prior results for specific
distance functions. For example, for the $\ell_1$ and $\ell_2^2$ case (and in general PSD matrices), [BCW20]
showed that a rank-$k$ approximation can be found in time $O(ndk/\varepsilon + nk^{w-1}/\varepsilon^{w-1})$. This bound
has extra $\text{poly}(1/\varepsilon)$ overhead compared to our bound stated in Table 2. The work of [IVWW19]
has a worse $\text{poly}(k, 1/\varepsilon)$ overhead for an additive error approximation for the $\ell_2$ case. See Section
1.2 for further discussion of prior works. The downstream applications of matrix-vector queries are
summarized in Table 2.

| Function | $f(x,y)$ | Preprocessing | Query Time | Reference |
|---|---|---|---|---|
| $\ell_p^p$ for $p$ even | $\|x-y\|_p^p$ | – | $O(ndp)$ | Thms. A.1 / A.3 |
| $\ell_p^p$ for $p$ odd | $\|x-y\|_p^p$ | $O(nd\log n)$ | $O(ndp)$ | Thms. 2.2 / A.4 |
| Mixed $\ell_\infty$ | $\max_{i,j}|x_i-y_j|$ | $O(nd\log n)$ | $O(n^2)$ | Thm. A.5 |
| Mahalanobis Distance$^2$ | $x^T M y$ | $O(nd^2)$ | $O(nd)$ | Thm. A.6 |
| Polynomial Kernel | $\langle x,y\rangle^p$ | – | $O(nd^p)$ | Thm. A.7 |
| Total Variation Distance | $\mathrm{TV}(x,y)$ | $O(nd\log n)$ | $O(nd)$ | Thm. A.8 |
| KL Divergence | $\mathrm{D_{KL}}(x\,\|\,y)$ | – | $O(nd)$ | Thm. A.2 |
| Symmetric Divergence | $\mathrm{D_{KL}}(x\,\|\,y)+\mathrm{D_{KL}}(y\,\|\,x)$ | – | $O(nd)$ | Thm. A.9 |
| Cross Entropy | $H(x,y)$ | – | $O(nd)$ | Thm. A.9 |
| Hellinger Distance$^2$ | $\sum_{i=1}^d \sqrt{x(i)y(i)}$ | – | $O(nd)$ | Thm. A.10 |

Table 1: A summary of our results for exact matrix-vector queries.

We also study fundamental limits for any upper bound algorithms. In particular, we show that *no algorithm* can compute a matrix-vector query for general inputs for the $\ell_\infty$ metric in subquadratic time, assuming a standard complexity-theory assumption called the *Strong Exponential Time Hypothesis (SETH)* [IP01, IPZ01].

**Theorem 1.2.** *For any $\alpha > 0$ and $d = \omega(\log n)$, any algorithm for exactly computing $Az$ for any input $z$, where $A$ is the $\ell_\infty$ distance matrix, requires $\Omega(n^{2-\alpha})$ time (assuming SETH).*

This shows a separation between the functions listed in Table 1 and the $\ell_\infty$ metric. Surprisingly, we can create queries for the *approximate* matrix-vector query in substantially faster time.

**Theorem 1.3.** *Suppose $X \subseteq \{0,1,\ldots,O(1)\}^d$. We can compute $By$ in time $O(n \cdot d^{O(\sqrt{d}\log(d/\varepsilon))})$ where $\|A-B\|_\infty \le \varepsilon$.*

To put the above result into context, the lower bound of Theorem 1.2 holds for points sets in $\{0,1,2\}^d$ in $d \approx \log n$ dimensions. In contrast, if we relax to an approximation guarantee, we can obtain a subquadratic-time algorithm for $d$ up to $\Theta(\log^2(n)/\log\log(n))$.

Finally, we provide a general understanding of the limits of our upper bound techniques. In Theorem B.1, we show that essentially the only $f$ for which our upper bound techniques apply have a "linear structure" after a suitable transformation. We refer to Appendix Section B for details.

> *2. We give algorithms for multiplying distance matrices faster than general matrices.*

Fast matrix-vector queries also automatically imply fast matrix multiplication, which can be reduced to a series of matrix-vector queries. For concreteness, if $f$ is the $\ell_p^p$ function which induces $A$, and $B$ is any $n \times n$ matrix, we can compute $AB$ in time $O(n^2 dp)$. This is substantially faster than the general matrix multiplication bound of $n^w \approx n^{2.37}$. We also give an improvement of this result for the case where we are multiplying two distance matrices arising from $\ell_2^2$. See Table 2 for summary.

> *3. We give fast algorithms for constructing distance matrices.*

Finally, we give fast algorithms for constructing approximate distance matrices. To establish some context, recall the classical Johnson-Lindenstrauss (JL) lemma which (roughly) states that a random projection of a dataset $X \subset \mathbb{R}^d$ of size $n$ onto a dimension of size $O(\log n)$ approximately preserves all pairwise distances [JL84]. A common applications of this lemma is to *instantiate* the $\ell_2$ distance matrix. A naive algorithm which computes the distance matrix after performing the JL projection requires approximately $O(n^2 \log n)$ time. Surprisingly, we show that the JL lemma is not tight with respect to creating an approximate $\ell_2$ distance matrix; we show that one can initialize the $\ell_2$ distance in an asymptotically better runtime.

**Theorem 1.4** (Informal; See Theorem D.5 ). *We can calculate a $n \times n$ matrix $B$ such that each $(i,j)$ entry $B_{ij}$ of $B$ satisfies $(1-\varepsilon)\|x_i-x_j\|_2 \le B_{ij} \le (1+\varepsilon)\|x_i-x_j\|_2$ in time $O(\varepsilon^{-2}n^2 \log^2(\varepsilon^{-1}\log n))$.*

| Problem | $f(x,y)$ | Runtime | Prior Work |
|---|---|---|---|
| $(1+\varepsilon)$ Relative error rank $k$ low-rank approximation | $\ell_1, \ell_2^2$ | $\tilde{O}\left(\frac{ndk}{\varepsilon^{1/3}} + \frac{nk^{w-1}}{\varepsilon^{(w-1)/3}}\right)$ Theorem C.4 | $O\left(\frac{ndk}{\varepsilon} + \frac{nk^{w-1}}{\varepsilon^{w-1}}\right)$ [BCW20] |
| Additive error $\varepsilon\|A\|_F$ rank $k$ low-rank approximation | $\ell_2$ | $\tilde{O}\left(\frac{ndk}{\varepsilon^{1/3}} + \frac{nk^{w-1}}{\varepsilon^{(w-1)/3}}\right)$ Theorem C.6 | $\tilde{O}(nd \cdot \mathrm{poly}(k, 1/\varepsilon))$ [IVWW19] |
| $(1+\varepsilon)$ Relative error rank $k$ low-rank approximation | Any in Table 1 | $\tilde{O}\left(\frac{Tk}{\varepsilon^{1/3}} + \frac{nk^{w-1}}{\varepsilon^{(w-1)/3}}\right)$ Theorem C.7 | $\tilde{O}\left(\frac{n^2dk}{\varepsilon^{1/3}} + \frac{nk^{w-1}}{\varepsilon^{(w-1)/3}}\right)$ [BCW22] |
| $(1 \pm \varepsilon)$ Approximation to top $k$ singular values | Any in Table 1 | $\tilde{O}\left(\frac{Tk}{\varepsilon^{1/2}} + \frac{nk^2}{\varepsilon} + \frac{k^3}{\varepsilon^{3/2}}\right)$ Theorem C.8 | $\tilde{O}\left(\frac{n^2dk}{\varepsilon^{1/2}} + \frac{nk^2}{\varepsilon} + \frac{k^{3\,3/2}}{\varepsilon}\right)$ [MM15] |
| Multiply distance matrix $A$ with any $B \in \mathbb{R}^{n \times n}$ | Any in Table 1 | $O(Tn)$ Lemma C.9 | $O(n^w)$ |
| Multiply two distance matrices $A$ and $B$ | $\ell_2^2$ | $O(n^2 d^{w-2})$ Lemma C.11 | $O(n^w)$ |

Table 2: Applications of our matrix-vector query results. $T$ denotes the matrix-vector query time, given in Table 1. $w \approx 2.37$ is the matrix multiplication constant [AW21].

Our result can be viewed as the natural runtime bound which would follow if the JL lemma implied an embedding dimension bound of $O(\mathrm{poly}(\log \log n))$. While this is impossible, as it would imply an exponential improvement over the JL bound which is tight [LN17], we achieve our speedup by carefully reusing distance calculations via tools from metric compression [IRW17]. Our results also extend to the $\ell_1$ distance matrix; see Theorem D.5 for details.

**Notation.** Our dataset will be the $n$ points $X = \{x_1, \dots, x_n\} \subset \mathbb{R}^d$. For points in $X$, we denote $x_i(j)$ to be the $j$th coordinate of point $x_i$ for clarity. For all other vectors $v$, $v_i$ denotes the $i$th coordinate. We are interested in matrices of the form $A_{i,j} = f(x_i, x_j)$ for $f : \mathbb{R}^d \times \mathbb{R}^d \to \mathbb{R}$ which measures the similarity between any pair of points. $f$ might not necessarily be a distance function but we use the terminology "distance function" for ease of notation. We will always explicitly state the function $f$ as needed. $w \approx 2.37$ denotes the matrix multiplication constant, i.e., the exponent of $n$ in the time required to compute the product of two $n \times n$ matrix [AW21].

## 1.2 Related Works

**Matrix-Vector Products Queries.** Our work can be understood as being part of a long line of classical works on the matrix free or implicit model as well as the active recent line of works on the matrix-vector query model. Many widely used linear algebraic algorithms such as the power method, the Lanczos algorithm [Lan50], conjugate gradient descent [S+94], and Wiedemann's coordinate recurrence algorithm [Wie86], to name a few, all fall into this paradigm. Recent works such as [MM15, BHSW20, BCW22] have succeeded in precisely nailing down the query complexity of these classical algorithms in addition to various other algorithmic tasks such as low-rank approximation [BCW22], trace estimation [MMMW21], and other linear-algebraic functions [SWYZ21b, RWZ20]. There is also a rich literature on query based algorithms in other contexts with the goal of minimizing the number of queries used. Examples include graph queries [Gol17], distribution queries [Can20], and constraint based queries [ES20] in property testing, inner product queries in compressed sensing [EK12], and quantum queries [LSZ21, CHL21].

Most prior works on query based models assume black-box access to matrix-vector queries. While this is a natural model which allows for the design non-trivial algorithms and lower bounds, it is not always clear how such queries can be initialized. In contrast, the focus of our work is not on obtaining query complexity bounds, but rather complementing prior works by creating an efficient matrix-vector query for a natural class of matrices.

**Subquadratic Algorithms for Distance Matrices.** Most work on subquadratic algorithms for distance matrices have focused on the problem of computing a low-rank approximation. [BW18, IVWW19] both obtain an additive error low-rank approximation applicable for all distance matrices. These works only assume access to the entries of the distance matrix whereas we assume we also have access to the underlying dataset. [BCW20] study the problem of computing the low-rank approximation of PSD matrices with also sample access to the entries of the matrix. Their results extend to low-rank approximation for the $\ell_1$ and $\ell_2^2$ distance matrices in addition to other more specialized metrics such as spherical metrics. Table 2 lists the runtime comparisons between their results and ours.

Practically, the algorithm of [IVWW19] is the easiest to implement and has outstanding empirical performance. We note that we can easily simulate their algorithm with no overall asymptotic runtime overhead using $O(\log n)$ vector queries. Indeed, their algorithm proceeds by sampling rows of the matrix according to their $\ell_2^2$ value and then post-processing these rows. The sampling probabilities only need to be accurate up to a factor of two. We can acquire these sampling probabilities by performing $O(\log n)$ matrix-vector queries which sketches the rows onto dimension $O(\log n)$ and preserves all row-norms up to a factor of two with high probability due to the Johnson-Lindenstrauss lemma [JL84]. This procedure only incurs an additional runtime of $O(T \log n)$ where $T$ is the time required to perform a matrix-vector query.

The paper [ILLP04] shows that the exact $L_1$ distance matrix can be created in time $O(n^{(w+3)/2}) \approx n^{2.69}$ in the case of $d = n$, which is asymptotically faster than the naive bound of $O(n^2 d) = O(n^3)$. In contrast, we focus on creating an (entry-wise) approximate distance matrices for all values of $d$.

We also compare to the paper of [ACSS20]. In summary, their main upper bounds are approximation algorithms while we mainly focus on exact algorithms. Concretely, they study matrix vector products for matrices of the form $A_{i,j} = f(\|x_i - x_j\|_2^2)$ for some function $f : \mathbb{R} \to \mathbb{R}$. They present results on approximating the matrix vector product of $A$ where the approximation error is additive. They also consider a wide range of $f$, including polynomials and other kernels, but the input to is always the $\ell_2$ distance squared. In contrast, we also present exact algorithms, i.e., with no approximation errors. For example one of our main upper bounds is an exact algorithm when $A_{i,j} = \|x_i - x_j\|_1$ (see Table 1 for the full list). Since it is possible to approximately embed the $\ell_1$ distance into $\ell_2^2$, their methods could be used to derive approximate algorithms for $\ell_1$, but not the exact ones. Furthermore, we also study a wide variety of other distance functions such as $\ell_\infty$ and $\ell_p^p$ (and others listed in Table 1) which are not studied in Alman et al. In terms of technique, the main upper bound technique of Alman et al. is to expand $f(\|x_i - x_j\|_2^2)$ and approximate the resulting quantity via a polynomial. This is related to our upper bound results for $\ell_p^p$ for even $p$ where we also use polynomials. However, our results are exact, while theirs are approximate. Our $\ell_1$ upper bound technique is orthogonal to the polynomial approximation techniques used in Alman et al. We also employ polynomial techniques to give upper bounds for the approximate $\ell_\infty$ distance function which is not studied in Alman et al. Lastly, Alman et al. also focus on the Laplacian matrix of the weighted graph represented by the distance matrix, such as spectral sparsification and Laplacian system solving. In contrast, we study different problems including low-rank approximations, eigenvalue estimation, and the task of initializing an approximate distance matrix. We do not consider the distance matrix as a graph or consider the associated Laplacian matrix.

It is also easy to verify the "folklore" fact that for a gram matrix $AA^T$, we can compute $AA^T v$ in time $O(nd)$ if $A \in \mathbb{R}^{n \times d}$ by computing $A^T v$ first and then $A(A^T v)$. Our upper bound for the $\ell_2^2$ function can be reduced to this folklore fact by noting that $\|x - y\|_2^2 = \|x\|_2^2 + \|y\|_2^2 - 2\langle x, y \rangle$. Thus the $\ell_2^2$ matrix can be decomposed into two rank one components due to the terms $\|x\|_2^2$ and $\|y\|_2^2$, and a gram matrix due to the term $\langle x, y \rangle$. This decomposition of the $\ell_2^2$ matrix is well-known (see Section 2 in [DPRV15]). Hence, a matrix-vector query for the $\ell_2^2$ matrix easily reduces to the gram matrix case. Nevertheless, we explicitly state the $\ell_2^2$ upper bound for completeness since we also consider all $\ell_p^p$ functions for any integer $p \geq 1$.

**Polynomial Kernels.** There have also been works on faster algorithms for approximating a kernel matrix $K$ defined as the $n \times n$ matrix with entries $K_{i,j} = k(x_i, x_j)$ for a kernel function $k$. Specifically for the polynomial kernel $k(x_i, x_j) = \langle x_i, x_j \rangle^p$, recent works such as [ANW14, AKK+20, WZ20, SWYZ21a] have shown how to find a sketch $K'$ of $K$ which approximately satisfies $\|K'z\|_2 \approx \|Kz\|_2$ for all $z$. In contrast, we can exactly simulate the matrix-vector product $Kz$. Our runtime is $O(nd^p)$ which has a linear dependence on $n$ but an exponential dependence on $p$ while the

aforementioned works have at least a quadratic dependence on $n$ but a polynomial dependence on $p$. Thus our results are mostly applicable to the setting where our dataset is large, i.e. $n \gg d$ and $p$ is a small constant. For example, $p = 2$ is a common choice in practice [CHC$^+$10]. Algorithms with polynomial dependence in $d$ and $p$ but quadratic dependence in $n$ are suited for smaller datasets which have very large $d$ and large $p$. Note that a large $p$ might arise if approximates a non-polynomial kernel using a polynomial kernel via a taylor expansion. We refer to the references within [ANW14, AKK$^+$20, WZ20, SWYZ21a] for additional related work. There is also work on kernel density estimation (KDE) data structures which upon query $y$, allow for estimation of the sum $\sum_{x \in X} k(x, y)$ in time sublinear in $|X|$ after some preprocessing on the dataset $X$. For widely used kernels such as the Gaussian and Laplacian kernels, KDE data structures were used in [BIMW21] to create a matrix-vector query algorithm for kernel matrices in time subquadratic in $|X|$ for input vectors which are entry wise non-negative. We refer the reader to [CS17, BCIS18, SRB$^+$19, BIW19, CKNS20] and references within for prior works on KDE data structures.

## 2 Faster Matrix-Vector Product Queries for $\ell_1$

We derive faster matrix-vector queries for distance matrices for a wide array of distance metrics. First we consider the case of the $\ell_1$ metric such that $A_{i,j} = f(x_i, x_j)$ where $f(x, y) = \|x - y\|_1 = \sum_{i=1}^{d} |x_i - y_i|$.

---

**Algorithm 1** Preprocessing

---

1: **Input:** Dataset $X \subset \mathbb{R}^d$
2: **procedure** PREPROCESSING
3:     **for** $i \in [d]$ **do**
4:         $T_i \leftarrow$ sorted array of the $i$th coordinates of all $x \in X$.
5:     **end for**
6: **end procedure**

---

---

**Algorithm 2** matrix-vector Query for $p = 1$

---

1: **Input:** Dataset $X \subset \mathbb{R}^d$
2: **Output:** $z = Ay$
3: **procedure** QUERY($\{T_i\}_{i \in [d]}, y$)
4:     $y_1, \cdots, y_n \leftarrow$ coordinates of $y$.
5:     Associate every $x_i \in X$ with the scalar $y_i$
6:     **for** $i \in [d]$ **do**
7:         Compute two arrays $B_i, C_i$ as follows.
8:         $B_i$ contains the partial sums of $y_j x_j(i)$ computed in the order induced by $T_i$
9:         $C_i$ contains the partial sums of $y_j$ computed in the order induced by $T_i$
10:     **end for**
11:     $z \leftarrow 0^n$
12:     **for** $k \in [n]$ **do**
13:         **for** $i \in [d]$ **do**
14:             $q \leftarrow$ position of $x_k(i)$ in the order of $T_i$
15:             $S_1 \leftarrow B_i[q]$
16:             $S_2 \leftarrow B_i[n] - B_i[q]$
17:             $S_3 \leftarrow C_i[q]$
18:             $S_4 \leftarrow C_i[n] - C_i[q]$
19:             $z(k) += x_k(i) \cdot (S_3 - S_4) + S_2 - S_1$
20:         **end for**
21:     **end for**
22: **end procedure**

---

We first analyze the correctness of Algorithm 2.

**Theorem 2.1.** *Let $A_{i,j} = \|x_i - x_j\|_1$. Algorithm 2 computes $Ay$ exactly.*

*Proof.* Consider any coordinate $k \in [n]$. We show that $(Ay)_k$ is computed exactly. We have

$$(Ay)(k) = \sum_{j=1}^{n} y_j \|x_k - x_j\|_1 = \sum_{j=1}^{n} \sum_{i=1}^{d} y_j |x_k(i) - x_j(i)| = \sum_{i=1}^{d} \sum_{j=1}^{n} y_j |x_k(i) - x_j(i)|.$$

Let $\pi^i$ denote the order of $[n]$ induced by $T_i$. We have

$$\sum_{i=1}^{d} \sum_{j=1}^{n} y_j |x_k(i) - x_j(i)| = \sum_{i=1}^{d} \left( \sum_{j:\pi^i(k) \leq \pi^i(j)} y_j (x_j(i) - x_k(i)) + \sum_{j:\pi^i(k) > \pi^i(j)} y_j (x_k(i) - x_j(i)) \right).$$

We now consider the inner sum. It rearranges to the following:

$$x_k(i) \left( \sum_{j:\pi^i(k) > \pi^i(j)} y_j - \sum_{j:\pi^i(k) \leq \pi^i(j)} y_j \right) + \sum_{j:\pi^i(k) \leq \pi^i(j)} y_j x_j(i) - \sum_{j:\pi^i(k) > \pi^i(j)} y_j x_j(i)$$
$$= x_k(i) \cdot (S_3 - S_4) + S_2 - S_1,$$

where $S_1, S_2, S_3$, and $S_4$ are defined in lines $15 - 18$ of Algorithm 2 and the last equality follows from the definition of the arrays $B_i$ and $C_i$. Summing over all $i \in [d]$ gives us the desired result. $\square$

The following theorem readily follows.

**Theorem 2.2.** *Suppose we are given a dataset $\{x_1, \ldots, x_n\}$ which implicitly defines the distance matrix $A_{i,j} = \|x_i - x_j\|_1$. Given a query $y \in \mathbb{R}^d$, we can compute $Ay$ exactly in $O(nd)$ query time. We also require a one time $O(nd \log n)$ preprocessing time which can be reused for all queries.*

## 3  Lower and Upper bounds for $\ell_\infty$

In this section we give a proof of Theorem 1.2. Specifically, we give a reduction from a so-called *Orthogonal Vector Problem* (OVP) [Wil05] to the problem of computing matrix-vector product $Az$, where $A_{i,j} = \|x_i - x_j\|_\infty$, for a given set of points $X = \{x_1, \ldots, x_n\}$. The orthogonal vector problem is defined as follows: given two sets of vectors $A = \{a^1, \ldots, a^n\}$ and $B = \{b^1, \ldots, b^n\}$, $A, B \subset \{0,1\}^d$, $|A| = |B| = n$, determine whether there exist $x \in A$ and $y \in B$ such that the dot product $x \cdot y = \sum_{j=1}^{d} x_j y_j$ (taken over reals) is equal to $0$. It is known that if OVP can be solved in strongly subquadratic time $O(n^{2-\alpha})$ for any constant $\alpha > 0$ and $d = \omega(\log n)$, then SETH is false. Thus, an efficient reduction from OVP to the matrix-vector product problem yields Theorem 1.2.

**Lemma 3.1.** *If the matrix-vector product problem for $\ell_\infty$ distance matrices induced by $n$ vectors of dimension $d$ can be solved in time $T(n, d)$, then OVP (with the same parameters) can be solved in time $O(T(n, d))$.*

*Proof.* Define two functions, $f, g : \{0,1\}^d \to [0,1]$, such that $f(0) = g(0) = 1/2$, $f(1) = 0$, $g(1) = 1$. We extend both functions to vectors by applying $f$ and $g$ coordinate wise and to sets by letting $f(\{a^1, \ldots, a^n\}) = \{f(a^1), \ldots, f(a^n)\})$; the function $g$ is extended in the same way for $B$. Observe that, for any pair of non-zero vectors $a, b \in \{0,1\}^d$, we have $\|f(a) - g(b)\|_\infty = 1$ if and only if $a \cdot b > 0$, and $\|f(a) - g(b)\|_\infty = 1/2$ otherwise.

Consider two sets of binary vectors $A$ and $B$. Without loss of generality we can assume that the vectors are non-zero, since otherwise the problem is trivial. Define three distance matrices: matrix $M_A$ defined by the set $f(A)$, matrix $M_B$ defined by the set $g(B)$ and $M_{AB}$ defined by the set $f(A) \cup f(B)$. Furthermore, let $M$ be the "cross-distance" matrix, such that such that $M_{i,j} = \|f(a^i) - g(b^j)\|_\infty$. Observe that the matrix $M_{AB}$ contains blocks $M_A$ and $M_B$ on its diagonal, and blocks $M$ and $M^T$ off-diagonal. Thus, $M_{AB} \cdot 1 = M_A \cdot 1 + M_B \cdot 1 + 2M \cdot 1$, where $1$ denotes an all-ones vector of the appropriate dimension. Since $M \cdot 1 = (M_{AB} \cdot 1 - M_A \cdot 1 - M_B \cdot 1)/2$, we can calculate $M \cdot 1$ in time $O(T(n, d))$. Since all entries of $M$ are either $1$ or $1/2$, we have that $M \cdot 1 < n^2$ if and only if there is an entry $M_{i,j} = 1/2$. However, this only occurs if $a^i \cdot b^j = 0$. $\square$

## 3.1 Approximate $\ell_\infty$ Matrix-Vector Queries

In light of the lower bounds given above, we consider initializing *approximate* matrix-vector queries for the $\ell_\infty$ function. Note that the lower bound holds for points in $\{0, 1, 2\}^d$ and thus it is natural to consider approximate upper bounds for the case of limited alphabet.

**Binary Case.** We first consider the case that all points $x \in X$ are from $\{0, 1\}^d$. We first claim the existence of a polynomial $T$ with the following properties. Indeed, the standard Chebyshev polynomials satisfy the following lemma, see e.g., see Chapter 2 in [SV$^+$14].

**Lemma 3.2.** *There exists a polynomial $T : \mathbb{R} \to \mathbb{R}$ of degree $O(\sqrt{d}\log(1/\varepsilon))$ such that $T(0) = 0$ and $|T(x) - 1| \leq \varepsilon$ for all $x \in [1/d, 1]$.*

Now note that $\|x - y\|_\infty$ can only take on two values, 0 or 1. Furthermore, $\|x - y\|_\infty = 0$ if and only if $\|x - y\|_2^2 = 0$ and $\|x - y\|_\infty = 1$ if and only if $\|x - y\|_2^2 \geq 1$. Therefore, $\|x - y\|_\infty = 0$ if and only if $T(\|x - y\|_2^2/d) = 0$ and $\|x - y\|_\infty = 1$ if and only if $|T(\|x - y\|_2^2/d) - 1| \leq \varepsilon$. Thus, we have that

$$|A_{i,j} - T(\|x_i - x_j\|_2^2/d)| = |\|x_i - x_j\|_\infty - T(\|x_i - x_j\|_2^2/d)| \leq \varepsilon$$

for all entries $A_{i,j}$ of $A$. Note that $T(\|x - y\|_2^2/d)$ is a polynomial with $O((2d)^t)$ monomials in the variables $x(1), \ldots, x(d)$. Consider the matrix $B$ satisfying $B_{i,j} = T(\|x_i - x_j\|_2^2/d)$. Using the same ideas as our upper bound results for $f(x, y) = \langle x, y \rangle^p$, it is straightforward to calculate the matrix vector product $By$ (see Section A.2). To summarize, for each $k \in [n]$, we write the $k$th coordinate of $By$ as a polynomial in the $d$ coordinates of $x_k$. This polynomial has $O((2d)^t)$ monomials and can be constructed in $O(n(2d)^t)$ time. Once constructed, we can evaluate the polynomial at $x_1, \ldots, x_n$ to obtain all the $n$ coordinates of $By$. Each evaluation requires $O((2d)^t)$ resulting in an overall time bound of $O(n(2d)^t)$.

**Theorem 3.3.** *Let $A_{i,j} = \|x_i - x_j\|_\infty$. We can compute $By$ in time $O(n(2d)^{\sqrt{d}\log(1/\varepsilon)})$ where $\|A - B\|_\infty \leq \varepsilon$.*

**Entries in $\{0, \ldots, M\}$.** We now consider the case that all points $x \in X$ are from $\{0, \ldots, M\}^d$. Our argument will be a generalization of the previous section. At a high level, our goal is to detect which of the $M + 1$ possible values in $\{0, \ldots, M\}$ is equal to the $\ell_\infty$ norm. To do so, we appeal to the prior section and design estimators which approximate the indicator function "$\|x - y\|_\infty \geq i$". By summing up these indicators, we can approximate $\|x - y\|_\infty$.

Our estimators will again be designed using the Chebyshev polynomials. To motivate them, suppose that we want to detect if $\|x - y\|_\infty \geq i$ or if $\|x - y\|_\infty < i$. In the first case, some entry in $x - y$ will have absolute value value at least $i$ where as in the other case, all entries of $x - y$ will be bounded by $i - 1$ in absolute value. Thus if we can boost this 'signal', we can apply a polynomial which performs thresholding to distinguish the two cases. This motivates considering the functions of $\|x - y\|_k^k$ for a larger power $k$. In particular, in the case that $\|x - y\|_\infty \geq i$, we have $\|x - y\|_k^k \geq i^k$ and otherwise, $\|x - y\|_k^k \leq di^{k-1}$. By setting $k \approx \log(d)$, the first value is much larger than the latter, which we can detect using the 'threshold' polynomials of the previous section.

We now formalize our intuition. It is known that appropriately scaled Chebyshev polynomials satisfy the following guarantees, see e.g., see Chapter 2 in [SV$^+$14].

**Lemma 3.4.** *There exists a polynomial $T : \mathbb{R} \to \mathbb{R}$ of degree $O(\sqrt{t}\log(t/\varepsilon))$ such that $|T(x)| \leq \varepsilon/t$ for all $x \in [0, 1/(10t)]$ and $|T(x) - 1| \leq \varepsilon/t^2$ for all $x \in [1/t, 1]$.*

Given $x, y \in \mathbb{R}^d$, our estimator will first try to detect if $\|x - y\|_\infty \geq i$. Let $T_1$ be a polynomial from Lemma 3.4 with $t = O(M^k)$ for $k = O(M\log(Md))$ and assuming $k$ is even. Let $T_2$ be a polynomial from Lemma 3.4 with $t = O(\sqrt{d}\log(M/\varepsilon))$. Our estimator will be

$$T_2\left(\frac{1}{d}\sum_{j=1}^{d} T_1\left(\frac{(x(j) - y(j))^k}{i^k \cdot M^k}\right)\right).$$

If coordinate $j$ is such that $|x(j) - y(j)| \geq i$, then

$$\frac{(x(j) - y(j))^k}{i^k \cdot M^k} \geq \frac{1}{M^k}$$

and so $T_1$ will evaluate to a value very close to 1. Otherwise, we know that

$$\frac{(x(j) - y(j))^k}{i^k \cdot M^k} \leq \frac{(i-1)^k}{i^k M^k} = \frac{1}{M^k} \left(1 - 1/i\right)^k \ll \frac{1}{M^k} \cdot \frac{1}{\text{poly}(M, d)}$$

by our choice of $k$, which means that $T_1$ will evaluate to a value close to 0. Formally,

$$\frac{1}{d} \sum_{j=1}^{d} T_1 \left( \frac{(x(j) - y(j))^k}{i^k \cdot M^k} \right)$$

will be at least $1/d$ if there is a $j \in [d]$ with $|x(j) - y(j)| \geq i$ and otherwise, will be at most $1/(10d)$. By our choice of $T_2$, the overall estimate output at least $1 - \varepsilon$ in the first case and a value at most $\varepsilon$ in the second case.

The polynomial which is the concatenation of $T_2$ and $T_1$ has $O\left( (dk \cdot \deg(T_1))^{\deg(T_2)} \right) = (dM)^{O(M\sqrt{d}\log(Md))}$ monomials, if we consider the expression as a polynomial in the variables $x(1), \ldots, x(d)$. Our final estimator will be the sum across all $i \geq 1$. Following our upper bound techniques for matrix-vector products for polynomial, e.g. in Section A.2, and as outlined in the prior section, we get the following overall query time:

**Theorem 3.5.** *Suppose we are given $X = \{x_1, \ldots, x_n\} \subseteq \{0, \ldots, M\}^d$ which implicitly defines the matrix $A_{i,j} = \|x_i - x_j\|_\infty$. For any query y, we can compute By in time $n \cdot (dM)^{O(M\sqrt{d}\log(Md/\varepsilon))}$ where $\|A - B\|_\infty \leq \varepsilon$.*

## 4 Empirical Evaluation

We perform an empirical evaluation of our matrix-vector query for the $\ell_1$ distance function. We chose to implement our $\ell_1$ upper bound since it's a clean algorithm which possesses many of the same underlying algorithmic ideas as some of our other upper bound results. We envision that similar empirical results hold for most of our upper bounds in Table 1. Furthermore, matrix-vector queries are the dominating subroutine in many key practical linear algebra algorithms such as the power method for eigenvalue estimation or iterative methods for linear regression: a fast matrix-vector query runtime automatically translates to faster algorithms for downstream applications.

| Dataset | $(n, d)$ | Algo. | Preprocessing Time | Avg. Query Time |
|---|---|---|---|---|
| Gaussian Mixture | $(5 \cdot 10^4, 50)$ | Naive | 453.7 s | 43.3 s |
| | | Ours | 0.55 s | 0.09 s |
| MNIST | $(5 \cdot 10^4, 784)$ | Naive | 2672.5 s | 38.6 s |
| | | Ours | 5.5 s | 1.9 s |
| Glove | $(1.2 \cdot 10^6, 50)$ | Naive | - | $\approx$ 2.6 days (estimated) |
| | | Ours | 16.8 s | 3.4 s |

Table 3: Dataset description and empirical results. $(n, d)$ denotes the number of points and dimension of the dataset, respectively. Query times are averaged over 10 trials with Gaussian vectors as queries.

**Experimental Design.** We chose two real and one synthetic dataset for our experiments. We have two "small" datasets and one "large" dataset. The two small datasets have $5 \cdot 10^4$ points whereas the large dataset has approximately $10^6$ points. The first dataset is points drawn from a mixture of three spherical Gaussians in $\mathbb{R}^{50}$. The second dataset is the standard MNIST dataset [LeC98] and finally, our large dataset is Glove word embeddings[2] in $\mathbb{R}^{50}$ [PSM14].

The two small datasets are small enough that one can feasibly initialize the full $n \times n$ distance matrix in memory in reasonable time. A $5 \cdot 10^4 \times 5 \cdot 10^4$ matrix with each entry stored using 32 bits requires 10 gigabytes of space. This is simply impossible for the Glove dataset as approximately 5.8 terabytes of space is required to initialize the distance matrix (in contrast, the dataset itself only requires $< 0.3$ gigabytes to store).

---

[2]Can be accessed here: http://github.com/erikbern/ann-benchmarks/

The naive algorithm for the small datasets is the following: we initialize the full distance matrix (which will count towards preprocessing), and then we use the full distance matrix to perform a matrix-vector query. Note that having the full matrix to perform a matrix-vector product only helps the naive algorithm since it can now take advantage of optimized linear algebra subroutines for matrix multiplication and does not need to explicitly calculate the matrix entries. Since we cannot initialize the full distance matrix for the large dataset, the naive algorithm in this case will compute the matrix-vector product in a standalone fashion by generating the entries of the distance matrix on the fly. We compare the naive algorithm to our Algorithms 1 and 2.

Our experiments are done in a 2021 M1 Macbook Pro with 32 gigabytes of RAM. We implement all algorithms in Python 3.9 using Numpy with Numba acceleration to speed up all algorithms whenever possible.

**Results.** Results are shown in Table 3. We show preprocessing and query time for both the naive and our algorithm in seconds. The query time is averaged over 10 trials using Gaussian vectors as queries. For the Glove dataset, it was infeasible to calculate even a single matrix-vector product, even using fast Numba accelerated code. We thus estimated the full query time by calculating the time on a subset of $5 \cdot 10^4$ points of the Glove dataset and extrapolating to the full dataset by multiplying the query time by $(n/(5 \cdot 10^4))^2$ where $n$ is the total number of points. We see that in all cases, our algorithm outperforms the naive algorithm in both preprocessing time and query time and the gains become increasingly substantial as the dataset size increases, as predicted by our theoretical results.

**Acknowledgements.** This research was supported by the NSF TRIPODS program (award DMS-2022448), Simons Investigator Award, MIT-IBM Watson AI Lab, GIST- MIT Research Collaboration grant, and NSF Graduate Research Fellowship under Grant No. 1745302.

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
