# OpenReview forum: "Faster Linear Algebra for Distance Matrices"
_NeurIPS.cc/2022/Conference — NeurIPS 2022 Accept_

### Official Review · Reviewer_37Ae · 2022-07-07

**Rating:** 8
**Confidence:** 4
**Soundness:** 4 excellent
**Presentation:** 2 fair
**Contribution:** 4 excellent

**Summary:**

The matrix-vector multiplication problem is fundamental and super important in ML applications. Faster algorithms for it are highly desirable.

This paper presents major speedups for the problem under the assumption that the matrix is not worst-case but is actually a "distance matrix" meaning that there is a set of n points x_1,...,x_n that implicitly define the matrix: the i,j entry is the distance between x_i and x_j.

This setting seems well-motivated and natural.

The complexity of this problem varies with the specific distance metric that is used, and with the dimension of the data-points x_1,..,x_n.

For \ell_1 distance over d dimensional points, the paper presents and algorithm computing the matrix-vector product in O(nd) time rather than the naive O(n^2) worst case bound. Technically, this algorithm is pretty simple and is at the level of an exercise.

There are other efficient algorithms for other norms that are not presented in the body of the paper.

For \ell_infinity over d=O(log n) dimensions, the paper shows a conditional lower bound stating that n^2 time is required, and then they present some interesting approximation algorithms. Both upper and lower bounds in this context are non-trivial but they use rather standard techniques in the context of nearest neighbor algorithms for such metrics.

**Questions:**

Are you aware of this paper? Isn't there some overlap in results/techniques:

"Algorithms and Hardness for Linear Algebra on Geometric Graphs"
Josh Alman; Timothy Chu; Aaron Schild; Zhao Song


**Strengths And Weaknesses:**

The results of this paper are super important and interesting, to a point that I would be surprised if they are actually new. But since I can't point to a previous reference that proves these results, I recommend accepting the paper.

The main weakness is that the techniques are not so groundbreaking; still, they are nontrivial observations that yield great results.

---

> ### Author Response · Authors · 2022-07-30
> **Response to reviewer 37Ae.**
>
> > The main weakness is that the techniques are not so groundbreaking; still, they are nontrivial observations that yield great results.
>
> We believe that the simplicity of our methods is a major strength as it allows for efficient and scalable implementations of our algorithms as demonstrated by our empirical results.
>
> > Are you aware of this paper? Isn't there some overlap in results/techniques:
> "Algorithms and Hardness for Linear Algebra on Geometric Graphs" Josh Alman; Timothy Chu; Aaron Schild; Zhao Song
>
> We thank the reviewer for bringing this relevant reference to our attention. In summary, their main upper bounds are approximation algorithms while we mainly focus on exact algorithms.
> Concretely, they study matrix vector products for matrices of the form $A_{i,j} = f(\|x_i - x_j\|_2^2)$ for some function $f: R \rightarrow R$. They present results on approximating the matrix vector product of $A$ where the approximation error is additive. They also consider a wide range of $f$, including polynomials and other kernels, but the input to $f$ is always the $\ell_2$ distance squared.
>
> In contrast, we present **exact** algorithms, i.e., with no approximation errors. For example one of our main upper bounds is an exact algorithm when $A_{i,j} = \| x_i - x_j \|_1$ (see Table 1 for the full list). Since it is possible to approximately embed the $\ell_1$ distance into $\ell_2^2$, their methods could be used to derive approximate algorithms for $\ell_1$, but not the exact ones. Furthermore, we also study a wide variety of other distance functions such as L-infinity and $\ell_p^p$  (and others listed in Table 1) which are not studied in Alman et al.
>
> In terms of technique, the main upper bound technique of Alman et al. is to expand $f(\|x_i - x_j\|_2^2)$ and approximate the resulting quantity via a polynomial. This is related to our upper bound results for the $\ell_p^p$ for $p$ even where we also use polynomials. However, our results are exact, while theirs are approximate. Our $\ell_1$ upper bound technique is orthogonal to the polynomial approximation techniques used in Alman et al. We also employ polynomial techniques to give upper bounds for the approximate L-infinity distance function which is not studied in Alman et al.
>
> Lastly, Alman et al. also focus on the Laplacian matrix of the weighted graph represented by the distance matrix, such as spectral sparsification and Laplacian system solving. In contrast, we study different problems including low-rank approximations, eigenvalue estimation, and the task of initializing an approximate distance matrix. We do not consider the distance matrix as a graph or consider the associated Laplacian matrix.  We will include a detailed comparison to this reference in the updated version of the paper.

---

### Official Review · Reviewer_hXqz · 2022-07-08

**Rating:** 7
**Confidence:** 3
**Soundness:** 4 excellent
**Presentation:** 2 fair
**Contribution:** 4 excellent

**Summary:**

This submission discusses 3 main results which the authors explicitly state on page 2 and again on pages 2 and 3.  Viz, for a distance matrix A of n points in d-dim Euclidean distance

o Computation of A.z in O(ndp)+ O(nd log(n)) steps for l_p norm 1<=p<inf
o Computation of A.B in O(n^2dp) steps for l_p norm 1<=p<=inf
o Computation of distance matrix A itself for n points in R^d in time O(n^2 log(n))

**Questions:**


I found Section B in the supplement to be fundamental to the rationale for writing this paper and its separation from the main text is incomprehensible except for the NeurIPS page limit, excluding supplementary material.

**Limitations:**

NA.

**Strengths And Weaknesses:**


This submission is effectively a long journal paper on numerical linear algebra divided in an ad hoc way to fit into the NeurIPS template.  The three results presented are significant and the proof methodology is intricate.  Efficacy of the results are well demonstrated by the experiments.

On the negative side, this paper has less relevance to ML and much more relevance to numerical linear algebra: its only claim for relevance to ML is that distance matrices arise in ML often as do need for the stated 3 results.  In this reviewer's opinion, this submission should be written in a linear way *not* separating material in the supplements from the main text (to meet NeurIPS page limit as in current form) thus presenting the material in a logical way in 30 or so pages and be submitted it to a numerical algebra publication (such as SIAM Journal in Numerical Algebra) where seasoned numerical analysts who only do matrix algebra will be able to provide much more valuable feedback to the authors.

---

> ### Author Response · Authors · 2022-07-30
> **Response to reviewer hXqz.**
>
> > On the negative side, this paper has less relevance to ML.
>
> We would like to highlight that many papers related to distance matrices have recently appeared in top machine learning conferences; for example, see the recent papers [1, 2, 3] which focus only on computing low-rank approximations of distance matrices.
>
> [1] Sublinear Time Low-Rank Approximation of Distance Matrices, Ainesh Bakshi, David Woodruff, NeurIPS 2018
>
> [2] Sample-Optimal Low-Rank Approximation of Distance Matrices, Piotr Indyk, Ali Vakilian and David Woodruff, COLT 2019
>
> [3] Sublinear Time Approximation of Text Similarity Matrices Archan Ray, Nicholas Monath, Andrew McCallum, Cameron Musco, AAAI 2022
>
> Furthermore, we believe that the wide applicability of distance function matrices in machine learning applications make this paper suitable for publication in NeurIPS. Indeed, measuring and representing pairwise similarities of points in a dataset is a fundamental ML primitive which is related to dimensionality reduction, learning manifolds, kernel methods, and many more applications.
>
> > I found Section B in the supplement to be fundamental to the rationale for writing this paper and its separation from the main text is incomprehensible except for the NeurIPS page limit, excluding supplementary material.
>
> We thank the reviewer for this valuable suggestion. If the paper is accepted, we will use the extra space allotted in the camera-ready version to move some of the discussions in Section B to the main body.

---

> > ### Author Response · Authors · 2022-08-07
> > **Follow up to reveiwer hXqz.**
> >
> > Dear Reviewer hXqz,
> >
> > Did we address all your concerns satisfactorily, namely your main concern about our paper's relevance to ML? If your concerns have not been resolved, could you please let us know which concerns were not sufficiently addressed so that we have a chance to respond?
> >
> > Many thanks,
> > The authors.

---

> > ### Comment · Reviewer_hXqz · 2022-08-07
> > **Rebuttal read**
> >
> > Thank you for your response.  I still believe linear algebra papers, even when relevant to ML applications, receive best reviews via professional linear algebra journals.  But your evidence of precedence is appreciated.

---

### Official Review · Reviewer_Ap29 · 2022-07-14

**Rating:** 8
**Confidence:** 4
**Soundness:** 4 excellent
**Presentation:** 4 excellent
**Contribution:** 3 good

**Summary:**

The authors propose fast algorithms for exact matrix-vector computations when the matrices are pairwise distance matrices (not strictly metrics). This leads to fast algorithms for matrix multiplication when one factor is a distance matrix.

They also give hopeless lower bounds for the $\ell_\infty$ distance, but fast approximate algorithms.

A third contribution is relegated to the appendix.

They present a small set of experiments to showcase the practical advantages of one of their algorithms.


**Questions:**

I'm a bit surprised that no one had come up with the algorithm for $\ell_1$ before. Could you perhaps comment on that?

**Limitations:**

The authors have adequately addressed the limitations and potential negative societal impact of their work.

**Strengths And Weaknesses:**

The paper is very well written and easy to follow. All the arguments in the main body seem sound to me.

One possible objection is the substance of the algorithms for $\ell_p$ distances, in particular $\ell_1$, which could almost be considered low-hanging fruit. The case of $\ell_\infty$ is much more inspired, and the reduction of OVP, while not very difficult, is non-trivial.

Minor: In the discussion of the binary case in section 3.1, t seems to come out of nowhere.

Some typos I ran into:
  - Theorem 1.1. z should be in R^n.
  - l. 162: "The sampling probabilities only needs"
  - l. 166: "T is the time requires to perform a matrix-vector query."
  - l. 271: "This motives"

---

> ### Author Response · Authors · 2022-07-30
> **Response to reviewer Ap29.**
>
> > I'm a bit surprised that no one had come up with the algorithm for ℓ1 before. Could you perhaps comment on that?
>
> To the best of our knowledge we do not believe exact matrix vector products for the $\ell_1$ distance function have been studied before. We asked several recognized experts in algorithms for linear algebra who also affirmed this.

---

### Author Response · Authors · 2022-07-30
**Thank you to reviewers.**

We thank the reviewers for their valuable feedback. Answers are given in a response to each review.

---

### Meta-Review · Area_Chair_wm8W · 2022-08-30

**Recommendation:** Accept
**Confidence:** Certain

**Metareview:**

The authors propose fast algorithms for exact matrix-vector computations when the matrices are pairwise distance matrices (not strictly metrics). This leads to fast algorithms for matrix multiplication when one factor is a distance matrix. Experimental results are included, as well as lower and upper bounds.

As the reviewers wrote, the paper is very well written and easy to follow. All the arguments in the main body seem sound.

I agree with the authors that this is a fundamental problem in machine learning with many modern applications.
Please consider adding some text from Section B as suggested by one of the reviewer.

**Award:**

No

---

### Decision · Program_Chairs · 2022-09-14

Accept